# Winning and losing in online gambling: Effects on within-session chasing

**Zhang Chen** [1]*, **Roos Arwen Doekemeijer**[1], **Xavier Noël**[2], **Frederick Verbruggen**[1]

**1** Department of Experimental Psychology, Ghent University, Ghent, Belgium, **2** Laboratoire de Psychologie Médicale et d'Addictologie, Faculté de Médecine, Université Libre de Bruxelles, Brussels, Belgium

* zhang.chen@ugent.be

**Data Availability Statement:** The anonymized per player summary data and the analysis code are publicly available at https://osf.io/vuy5d/. All reported results can be reproduced with the summary data.

## Abstract

The tendency to continue or intensify gambling after losing (loss-chasing) is widely regarded as a defining feature of gambling disorder. However, loss-chasing in real gambling contexts is multifaceted, and some aspects are better understood than others. Gamblers may chase losses between multiple sessions or within a single session. Furthermore, within a session, loss-chasing can be expressed in the decision of (1) when to stop, (2) how much stake to bet, and (3) the speed of play after winning and losing. Using a large player-tracking data set (>2500 players, >10 million rounds) collected from the online commercial game *Mystery Arena*, we examined these three behavioral expressions of within-session loss-chasing. While the first two aspects (when to stop and how much stake to bet) have been examined previously, the current research is the first large-scale study to examine the effects of wins and losses on the speed of play in real gambling. The players were additionally assigned different involvement levels by the operator based on their gambling behavior on the operator's own platform, which further allowed us to examine group differences in loss-chasing. We found that after winning, both the high- and low-involvement groups were less likely to stop, and increased the stake amount, thus showing win-chasing instead of loss-chasing in these two facets. After losing, both groups played more quickly though, which may reflect an urge to continue gambling (as an expression of loss-chasing). Wins and losses had a smaller influence on the speed of play for the high-involvement players, suggesting that they might have reduced sensitivity to wins and/or losses. Future work can further examine chasing in different gambling products and in people with gambling problems to assess the generalizability of these findings.

## Introduction

Gambling is a popular activity worldwide. While most people gamble within their means, for a small yet significant proportion of the population, it may spiral out of control and develop into problem gambling [1, 2] (the past-year problem gambling rates vary between 0.1% and 5.8% worldwide [3]). Excessive gambling can lead to adverse consequences for the people with gambling problems, those around them, and the society at large [4].

**Funding:** This work was supported by an ERC Consolidator grant (European Union's Horizon 2020 research and innovation programme, grant agreement No 769595; https://erc.europa.eu/) and Methusalem Project 01M00221 (Ghent University; https://www.ugent.be/en/research/funding/bof/methusalem) awarded to Frederick Verbruggen. Zhang Chen is supported by a Postdoctoral fellowship of the Scientific Research Foundation, Flanders (FWO-Vlaanderen; https://www.fwo.be/ Xavier Noël is supported by the Fonds de la Recherche Scientifique (F.R.S.-FNRS; https://www.frs-fnrs.be/en/), Belgium and research funding from BAGO (Belgian Association of Gaming Operators; https://bago.be/). The funders had no role in study design, data collection and analysis, decision to publish, or preparation of the manuscript.

**Competing interests:** I have read the journal's policy and the authors of this manuscript have the following competing interests: ZC, RD, and FV declare that no competing interests exist. XN received research funding from BAGO (Belgian Association of Gaming Operators) of which Gaming1 is affiliated. No honorarium was received. There was no editorial direction or censorship from Gaming1. This does not alter our adherence to PLOS ONE policies on sharing data and materials.

One important behavioral phenomenon that is generally considered to mark the transition from non-problem to problem gambling is chasing [5]. 'Chasing', or more commonly loss-chasing, describes a gambler's tendency to continue or intensify gambling to recoup previous losses. Loss-chasing is widely regarded as a defining feature of gambling disorder [6, 7]. Further understanding of whether and how gamblers chase losses (across the whole spectrum of gambling involvement) will therefore provide important insights into the development of gambling problems. Such knowledge will also be crucial in developing tools that can effectively detect and reduce problem gambling.

Loss-chasing can be expressed in multiple aspects of gambling behavior. Players may chase losses *between sessions*, such as returning another time to recover previous losses, or *within sessions*, by continuing or intensifying gambling after losses within a session [8]. In this paper, we focus on within-session loss-chasing in an online commercial gambling product (with behavioral tracking data from >2500 players and >10 million rounds). We examined three purported behavioral expressions of within-session chasing, namely (1) when players decided to end a session, (2) how much stake they placed in each round, and (3) how quickly they played. Most previous studies on chasing have focused on the first two aspects, while the speed of play as a potential expression of chasing has received less attention (certainly in real gambling games). The current research is the first large-scale study to examine the effects of wins and losses on the speed of play in real gambling. Below we introduce the three facets of within-session loss-chasing in turn.

## Three expressions of within-session loss-chasing

Gamblers may chase losses by continuing a gambling session after a loss. To test how wins and losses influence the decision of when to stop, Leino and colleagues examined when players ended a session when playing real electronic gaming machines (EGMs) [9]. They found that EGM players were more likely to end a session after losing than after winning, thus showing no loss-chasing in this facet. One explanation may be that the EGM players in Leino et al.'s study were mostly people without gambling problems, who were able to stop after losing to avoid further losses. Interestingly though, some work has shown that people with gambling problems also did not show loss-chasing in this aspect. For instance, three studies found that participants with greater problem gambling severity were more likely to play beyond a required number of rounds, and played more rounds in total in simulated gambling, compared to those without gambling problems [10–12]. Intriguingly, in all three studies, the nominal win or loss in the preceding 30 rounds did not influence participants' decision of when to stop. In other words, people with gambling problems played longer sessions in general, but the decision of when to end a session was not influenced by wins or losses per se.

People may also chase losses by placing a higher stake, sometimes beyond the amount they initially intended to spend [8, 13]. While some previous work has indeed found increased stakes after a loss [14–16], others have observed the opposite [17, 18]. Whether people bet larger or smaller amounts after losing may depend on many factors, such as whether the gamble provides the opportunity to offset prior losses (i.e., the break-even effect [17]), the presentation format of the gamble [19], and whether the losses are realized or not [20]. The last finding, namely the realization effect, may be especially relevant here. Imas [20] showed in a lab-based study that when losses were realized, such as when money was transferred between accounts after losing, people took less 'risk' by making a smaller bet after a loss; in contrast, when losses were unrealized, such as when the loss was not converted into real money yet, people took more 'risk' by making a larger bet after losing. In real online

gambling, whether players will place a higher or a lower stake after losing may depend on whether they perceive the loss to be 'realized' or not. On the one hand, the loss may be seen as 'realized', as money is taken away from players' accounts. On the other hand, the loss may still be 'unrealized', because the money on the gambling platform may be perceived as game token (i.e., play money) rather than real money. We will therefore examine this facet in the current study.

Most previous research has focused on the decision of when to stop or how much money to bet as expressions of chasing. Here we argue that the speed of play might also be a crucial aspect of chasing. Previous work has shown that in simulated gambling or gambling-like tasks in the laboratory, participants initiated a new game more quickly after a loss than after a win [21–24] and a non-gamble trial [25, 26]. The faster initiation of a new gamble after a loss (the so-called post-loss speeding effect [25]) may reflect a stronger urge to continue playing after losing, and can therefore be seen as an expression of loss-chasing [13]. Speeding up after losing may lead to a vicious circle between continued gambling and losing, as games of faster speed of play are generally experienced as more exciting and harder to stop, especially for people with gambling problems [27–29]. It is therefore important to examine how wins and losses may influence the speed of play in real (online) gambling.

## The current research

With regards to the three behavioral expressions of within-session loss-chasing discussed above, people seemed to chase losses in the speed of play, but not in the decision of when to end a session. In terms of how much money to bet, inconsistent findings have been reported. However, it is important to note that loss-chasing is a multifaceted and highly dynamic construct. The behavioral expressions of loss-chasing may vary across different gambling contexts (e.g., simulated gambling in the lab vs. real gambling), different types of gambling games (e.g., casino games vs. poker games [30]), and different subgroups of gamblers (e.g., people with different levels of gambling involvement). Much previous work on loss-chasing is conducted in laboratory contexts. While such studies have provided valuable insights into loss-chasing, it is also important to systematically characterize whether and how gamblers may chase losses in real gambling contexts.

In the current research, we examined how wins and losses might influence the three expressions of within-session chasing in an online commercial gambling product *Mystery Arena*, with a large behavioral tracking data set. Given the multifaceted and dynamic nature of loss-chasing and some inconsistent findings in the literature, *a priori* we did not have strong predictions for whether loss-chasing would occur in each facet or not. This study was therefore exploratory in nature, and mainly aimed at characterizing loss-chasing (i.e., its presence or absence in each facet) in real online gambling. Note though that we selected a gambling product that seemed most suitable for examining the speed of play, as this aspect of gambling behavior is often ignored.

The large number of players additionally allowed us to explore how loss-chasing may vary as a function of gambling involvement. People differ in how involved they are in gambling, with those with gambling problems constituting a subgroup that has a very high level of involvement. Examining loss-chasing across the whole spectrum will provide insights into the role of loss-chasing in the progression of gambling involvement. The gambling operator gathered data on each player's gambling involvement on their platform (across all games), based on the players' money and time expenditure. We therefore used this data to examine how the three expressions of loss-chasing may differ between gamblers with high versus low levels of involvement.

## Materials and methods

### Participants

We analyzed data from *Mystery Arena*, developed by Gaming1. In total, 2713 players were randomly selected by the operator and data on their play behavior were retrieved in March 2020. The retrieved data covered a period since 2014.

### Procedure

Below we first describe the general rules of the game, and then explain how players interacted with the game interface in more detail.

**General rules of the game.** Before starting a round, players needed to place a stake (Fig 1). The stake was for the whole round, and could not be changed after pressing start. Twelve columns of dice (three dice per column) then appeared one by one. Players needed to put these 12 columns into 4 slots, with 3 columns in each slot. They won points in a slot when a horizontal or a diagonal line of the slot contained the same dice. After all 12 columns had been placed, the points from all 4 slots were added up and converted into monetary prizes (win or loss). After finishing a round, players could continue or end the session.

Gambling products are often categorized into two broad types, chance-based games that involve little deliberation or skill (e.g., slot machines, lotteries), and skill-based games in which players can exert some influence over the outcome (e.g., Blackjack, sports betting) [31]. On the continuum from chance-based to skill-based games, *Mystery Arena* sits between the two endpoints, but probably somewhere closer to the endpoint of chance-based games. That is, the dice are randomly generated, which largely determines the outcome (i.e., the chance component). However, players can decide in which slot to place each column, which also influences the outcome (i.e., the skill component).

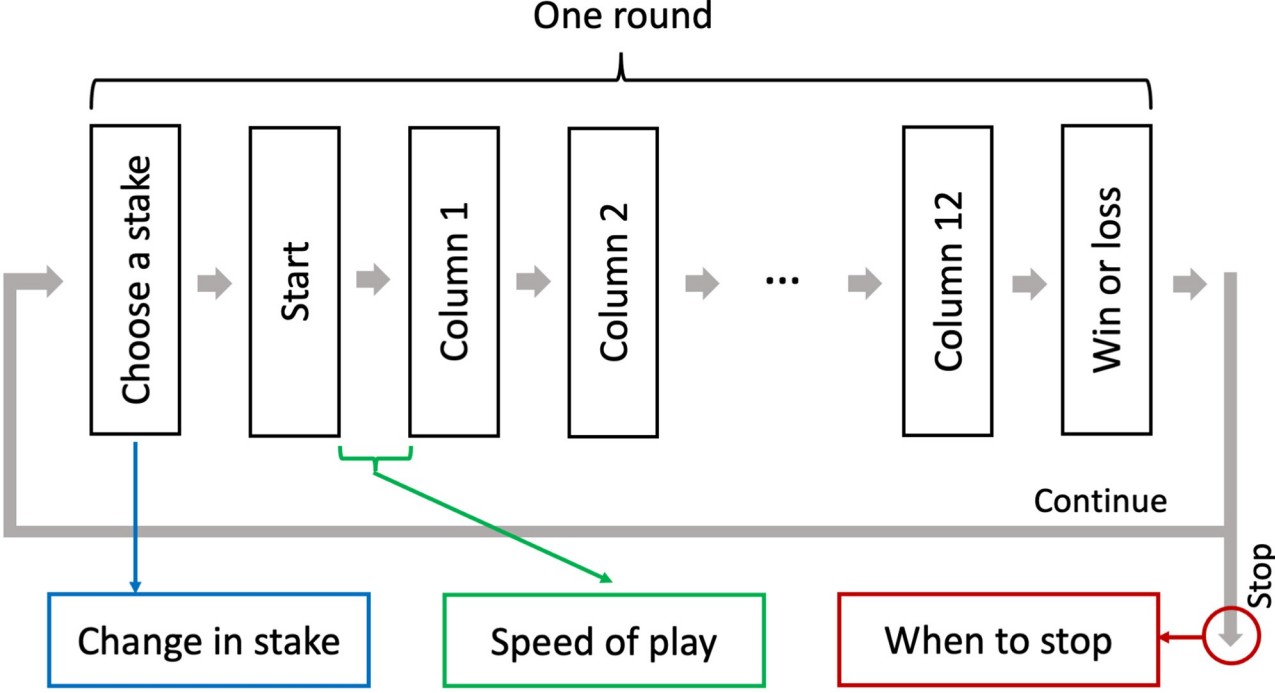

**Fig 1. A flowchart showing the events within one round, and on which event each expression of loss-chasing is based.**

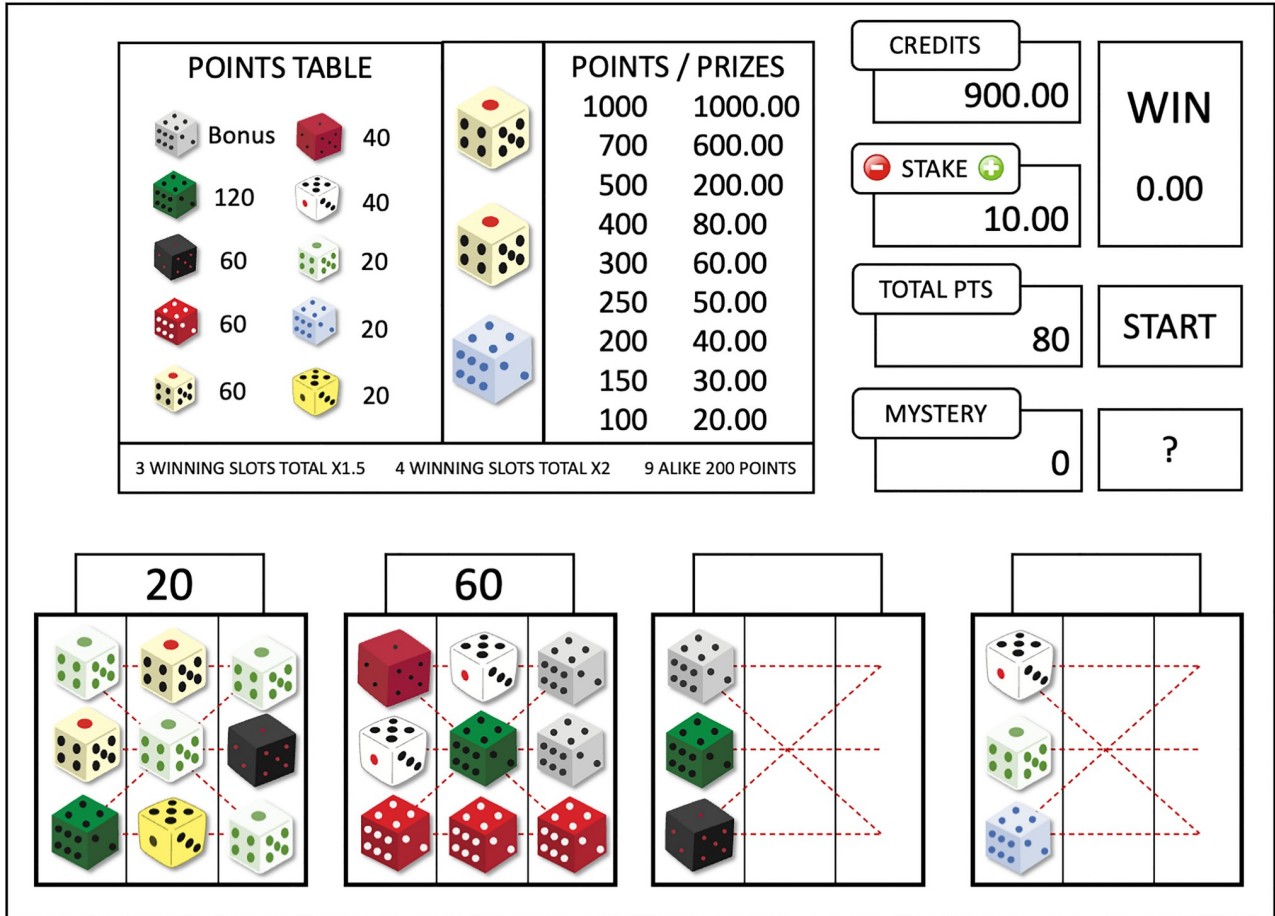

**Fig 2. A schematic illustration of the game interface.** In each round, players place 12 columns of dice (three dice per column) into 4 slots one by one. They win points in a slot if a horizontal or a diagonal line of the slot contains the same dice. The points from all 4 slots are added up and converted into monetary prizes (win or loss).

**The game interface.** Fig 2 provides a schematic illustration of the game interface. Before starting a round, players could adjust the stake (from 0.25 euro to 20 euro, ten levels in total) by clicking on the plus and the minus button on the "STAKE" panel. As they adjusted the stake, the monetary prizes were adjusted accordingly in the "POINTS / PRIZES" panel. The stake acted as a multiplier, changing the monetary prizes paired with the points. After choosing a stake, players started the game by clicking on the "START" button, or anywhere except the "STAKE" panel.

A column of three dice was then shown (between the "POINTS TABLE" panel and the "POINTS / PRIZES" panel, see Fig 2). There were 10 different dice, with each dice worth a certain amount of points or a bonus play. Each dice was randomly drawn with equal probabilities. Four slots were displayed on the lower half of the interface. Players could place the column of dice into one of the four slots by clicking on the slot. After a column was placed, a new column of three dice was shown. Each slot was filled from left to right. If any of the three horizontal lines or the two diagonal lines in a slot contained the same dice, participants received corresponding points. After all twelve columns were placed, the points won from all four slots were summed up. The sum was multiplied by 1.5 if players won points in 3 slots, multiplied by 2 if they won points in all 4 slots, and players received an extra 200 points if the 9 dices in any slot

were the same. In case of a win ($\geq$ 100 points), the points were converted to a monetary prize, and displayed in the 'WIN' panel. The monetary prize was then reduced to 0 in the 'WIN' panel while the available credits increased (i.e., 'transferring' money into players' accounts), along with an animation and sounds of pouring golden coins. At the end of the animation, the message 'PLAY AGAIN' appeared. In case of a loss ($<$ 100 points), players directly got the message 'PLAY AGAIN'. Wins and losses thus differed in their visual and auditory feedback events. In both cases, players could advance to the next round by clicking on the 'START' button or anywhere except the 'STAKE' panel. Note that players could also start a new round *before* the 'PLAY AGAIN' message appeared. Due to the differences in the feedback after wins and losses, and the flexibility in when to start a new round, we did not use the speed of starting a new round as the indicator for the speed of play. Instead, we used the interval between starting a new round till placing the first column in the new round (see Fig 1), which is not confounded by the factors discussed above.

If a certain slot contained three dice for a bonus play (on a horizontal or a diagonal line), the main game would pause and players would play a bonus game to win extra rewards. After the bonus game, the main game resumed. The bonus game occurred in 1.47% of the rounds in the current data. We do not have data on players' responses in these bonus games. In the analysis, we included all rounds both with and without bonus games. In other words, we analyzed the effects of winning and losing on chasing behavior, regardless of whether players won from the main game or the bonus game. Furthermore, in the main game (with the 4 slots), instead of using the computer mouse, players could use the space bar or the Enter key on the keyboard to play. In that case, the program automatically determined the best location to place each column. This would presumably allow players to play at a higher speed than using the computer mouse. Players may also use mobile devices instead of desktops to play [32]. We do not have data on whether players used desktops or mobile devices, and whether they used the computer mouse or the keyboard to play.

## Measures

**Behavioral tracking data.** Table 1 provides an overview of the variables in the behavioral tracking data, and the explanation for each variable.

**Defining the low- and high-involvement groups.** The operator also tracks each player's overall monetary and time expenditure on their platform (see S1 File for a detailed description

**Table 1. An overview of the variables in the behavioral tracking data.**

| Variable | Explanation |
| --- | --- |
| Player ID | A unique, deidentified ID for each player. |
| Session ID | A unique, deidentified ID for each session. One session consists of all rounds from when a player logged into the game, till when they disconnected. |
| Round ID | A unique, deidentified ID for each round. One round consists of placing all 12 columns of dice into the 4 slots. |
| Stake amount | The stake amount in each round, in euro. |
| Win amount | The amount of money a player won in a round, in euro. In case of a loss, the win amount is 0. |
| Speed of starting a round | The duration from when players placed the last column in the previous round, till when they started the current round, in milliseconds. |
| Speed of placing the columns | How quickly players placed each of the 12 columns into the slots, in milliseconds. Note that we used the speed of placing the first column as the behavioral indicator of speed of play. |
| Bonus game | Whether a bonus game occurred or not in a round. |

of the indicators). Based on these indicators, each player is assigned a level of gambling involvement from the operator, from 0 to 5, with a higher level standing for higher involvement. Gaming1 provided us with data from players with levels 0, 3, 4, and 5, but no data from players with levels 1 and 2. Since the indicators that players exhibit may vary over time, the involvement level of a player can accordingly also change. This is the case for some level 3/4/5 players in the data, where one player had different involvement levels across time. All level 0 players in the data had consistently level 0. Because the involvement level varied within some players, and that we do not have data on which indicators each player exhibited (i.e., Gaming1 only provided us general scores), we therefore focused on the comparison between the level 0 players and the level 3/4/5 players, without making further distinctions among levels 3, 4 and 5. We will refer to the level 0 players as the low-involvement group, and the level 3/4/5 players as the high-involvement group.

## Statistical analysis

Data analyses were conducted in R (4.0.2) [33], using R packages afex (0.27–2) [34], BayesFactor (0.9.12–4.2) [35], bookdown (0.21) [36], ggplot2 (3.3.2) [37], ggpubr (0.4.0) [38], kableExtra (1.2.1) [39], knitr (1.29) [40], Rmisc (1.5) [41], and tidyverse (1.3.0) [42].

**Players and play behavior.** To examine whether and to what extent the two groups differed in their gambling involvement, we conducted a series of comparisons between the high- and low-involvement groups (see the 'Defining the low- and high-involvement groups' section above for how the two groups are defined). We used both the frequentist Welch's t-tests [43] and Bayesian t-tests. For all Bayesian t-tests, a Cauchy distribution with a scale parameter of 0.707 (the default in the BayesFactor package) was used as the prior. Bayesian t-tests provide Bayes factors, which indicate the relative likelihood of obtaining the observed data under the alternative hypothesis versus the null hypothesis [44]. We reported the natural logarithm of Bayes factors (lnBF), as some Bayes factors were large. To interpret the Bayes factors, we used the widely adopted verbal labels [45]. For instance, a lnBF above 3.40 (i.e., a BF above 30) is considered 'very strong' evidence for the alternative hypothesis. P values were corrected for multiple comparisons using the Holm-Bonferroni method. Hedges' *g* was reported as effect sizes for t tests [46].

**Overview of the analyses on loss-chasing.** We conducted three sets of analyses, each focusing on one facet of loss-chasing. Table 2 provides an overview of the behavioral indicator used, the main analysis, the expression of loss-chasing, and (as a preview of the results) whether loss-chasing was observed or not for each facet. In the sections that follow, we explain each set of analyses in detail.

**Table 2. An overview of the behavioral indicator, main analysis, loss-chasing expression and whether it was observed in the current study for each facet.**

| Facet | Behavioral indicator | Main analysis | Loss-chasing expression | Observed? |
|---|---|---|---|---|
| When to stop | The probabilities of ending a session after winning and losing, while controlling for the overall probability of stopping. | Mixed ANOVA on the relative likelihoods of stopping, with the prior outcome (loss vs. win, within-subjects) and involvement level (high vs. low, between-subjects) as factors. | The probability of ending a session will be lower after a loss than after a win. | No |
| Change in stake | The probability of changing the stake, and the average change in stake amount after winning and losing. | Mixed ANOVA on the probability of changing stake and the average change in stake amount, with the prior outcome (loss vs. win, within-subjects) and involvement level (high vs. low, between-subjects) as factors. | Players will increase the stake amount more after a loss than after a win. | No |
| Speed of play | How quickly players place the first column of dice after starting a round (z score) after winning and losing. | Mixed ANOVA on the mean RT z scores, with the prior outcome (loss vs. win, within-subjects) and involvement level (high vs. low, between-subjects) as factors. | Players will place the first column of a round more quickly after a loss than after a win. | Yes |

**When to stop.** *Aim of analyses*. The first set of analyses focused on whether winning and losing would influence the probability of ending a session, as an expression of loss-chasing.

*Behavioral indicator*. We used the probabilities of ending a session after a win and after a loss as the behavioral indicators of when to stop. More concretely, for each player, we computed the total number of winning rounds (all sessions combined), and the number of winning rounds that were at the end of a session (i.e., players decided to stop after these rounds). Dividing the latter by the former resulted in the probability of stopping after winning (i.e., p(*stop|win*)). The probability of stopping after losing (i.e., p(*stop|loss*)) was then similarly computed. If players chased losses by continuing playing, p(*stop|loss*) would be lower than p(*stop|win*). Note that this analysis is conceptually similar to the logistic regression used by Leino et al. [9], who examined the effects of gambling outcomes on the decision to continue vs. stop as a binary variable.

Importantly though, since the high-involvement players on average had longer sessions (as a criterion to create the two groups), the overall probability of ending a session was lower for the high-involvement players than for the low-involvement players (see S1 File). We therefore had to control for the overall probability of stopping in the analyses. For each player, we computed the total number of rounds (all sessions combined, regardless of the outcome) and the number of rounds at the end of a session. Dividing the latter by the former resulted in the overall probability of stopping (i.e., p(*stop − overall*)). For each player, we then divided both p(*stop|win*) and p(*stop|loss*) by p(*stop − overall*). The resulting values indicated the relative likelihoods of stopping after winning and losing, controlling for each player's general probability of stopping.

*Analysis methods*. To achieve reliable estimates of p(*stop|win*) and p(*stop|loss*), players needed to have at least 5 wins and 5 losses in the data. For the included players, the relative likelihoods of stopping were computed as outlined above, and analyzed using a mixed ANOVA with prior outcome (loss vs. win, within-subjects) and involvement level (high vs. low, between-subjects) as independent variables. We then conducted a series of within-group and between-group pairwise comparisons to break down the effects from ANOVA.

**Change in stake.** *Aim of analyses*. The second set of analyses examined whether players would chase losses within a session by increasing the stake amount more after losing than after winning.

*Behavioral indicator*. We used two behavioral indicators, namely the probability of changing the stake, and the average change in stake amount, for after winning and losing respectively. High-involvement players overall placed higher stakes than low-involvement players (again, as a criterion to create the two groups). We therefore used the change in stake rather than the absolute stake amount as the behavioral indicator. More concretely, for each round, we computed the difference between the current stake amount and the previous stake amount. Average changes in stake amount were then computed for after a win and after a loss, separately.

*Analysis methods*. The first round of each session was excluded, as it did not have a previous round. To achieve reliable estimates, players needed to have at least 5 rounds following a win and 5 rounds following a loss. For each player, we calculated the probability of changing stake and the average change in stake amount (in euro cents) after a win and a loss. A positive (negative) change in stake amount means players increased (decreased) the stake in the current round. The probability of changing stake and the change in stake amount (euro cents) were submitted to a mixed ANOVA, with prior outcome (loss vs. win, within-subjects) and involvement level (high vs. low, between-subjects) as independent variables.

**Speed of play.** *Aim of analyses*. The last set of analyses examined whether players would chase losses by playing more quickly after losing than after winning.

*Behavioral indicator*. The speed of starting a round in the behavioral tracking data is the duration from when players placed the last column in the *previous* round till when they started the current round (Table 1). This duration included the different feedback events after winning and losing, and the time spent on adjusting the stake. Due to these confounding factors, we used the response time (RT) of placing the first column as the behavioral indicator of speed of play instead. This RT is from when players started the current round till when they put the first column in one of the 4 slots, and thus is not confounded by the factors mentioned above.

The high-involvement players overall played more quickly than the low-involvement players (see the Supplemental Materials), which might attenuate the influence of wins and losses on the speed of play. To control for the general playing speed, we therefore standardized the RTs within each player (i.e., RT z scores = (RT—mean RT of each player) / standard deviation of RTs for each player), and used the RT z scores in the analyses.

*Analysis methods*. We used the same RT exclusion criteria as in previous work [25, 26]. The first round of each session (i.e., no prior outcome) and rounds where the RT was above 5000 milliseconds were excluded. Players needed to have at least 5 rounds after a loss and 5 rounds after a win to be included in the analysis. The RT data were then standardized within each player. The mean RT z scores were calculated for each player, for rounds after a win and after a loss, respectively. We again used a mixed ANOVA, with prior outcome (loss vs. win, within-subjects) and involvement level (high vs. low, between-subjects) as independent variables.

## Ethics

The study procedures were carried out in accordance with the Declaration of Helsinki. When creating an account on the operator's platform, players confirmed (by clicking on a checkbox) that they had read the terms and conditions and the privacy policy, and thus provided consent for the use of their data for research purposes. Gaming1 provided the deidentified data to the researchers for statistical analysis. The current project was approved by the ethics committee of the faculty of psychology at the Université Libre de Bruxelles. Gaming1 does not allow the round-level raw data to be shared. However, the anonymized per player summary data and the analysis code are publicly available at https://osf.io/vuy5d/. All reported results can be reproduced with the summary data.

## Results

### Players and play behavior

**Players.** In total, the sample consisted of 2713 players, with 910 players in the low-involvement group (i.e., level 0; 409 females, 501 males, $M_{age}$ = 37.7, $SD_{age}$ = 12.7) and 1803 players in the high-involvement group (i.e., levels 3, 4, or 5; 800 females, 1003 males, $M_{age}$ = 40.2, $SD_{age}$ = 10.9). See S1 File for distributions of age and gender, and the total number of sessions and rounds played for both groups.

**Play behavior.** In total, players played 120,863 sessions and 10,370,013 rounds. We compared the following parameters between the two groups (see Table 3): (1) the total number of sessions played per player, (2) the total number of rounds played per player, (3) the mean number of rounds per session, (4) the median number of rounds per session, (5) the mean stake (in euro) in each round, (6) the median stake (in euro) in each round, (7) the probability of winning in all rounds combined, (8) the mean win amount (the presented win amount minus the stake, in euro) per round, in the rounds where players won, (9) the median win amount (in euro) per round, in the rounds where players won, (10) the mean loss amount (the loss amount equals the stake, in euro), in the rounds where players lost, (11) the median loss

**Table 3. Comparing play behavior between the two groups.**

| Parameter | High (N = 1803) | Low (N = 910) | diff | lowerCI | upperCI | df | t | p | lnBF | g |
|---|---|---|---|---|---|---|---|---|---|---|
| (1) Session number | 64.2 (162.8) | 5.6 (12.2) | 58.6 | 51.1 | 66.2 | 1842.2 | 15.2 | <.001 | 54.3 | 0.618 |
| (2) Round number | 5590.3 (13764) | 319.5 (833.7) | 5270.7 | 4632.6 | 5908.8 | 1828.1 | 16.2 | <.001 | 61.6 | 0.659 |
| (3) Mean round number | 82.1 (78.3) | 51.3 (63.3) | 30.8 | 25.4 | 36.3 | 2195.6 | 11.0 | <.001 | 48.7 | 0.449 |
| (4) Median round number | 60.6 (65.4) | 45.9 (60.2) | 14.8 | 9.8 | 19.7 | 1963.6 | 5.8 | <.001 | 12.9 | 0.238 |
| (5) Mean stake (Euro) | 2.2 (3.2) | 0.7 (0.9) | 1.5 | 1.3 | 1.6 | 2344.2 | 18.2 | <.001 | 87.0 | 0.738 |
| (6) Median stake (Euro) | 1.9 (3.3) | 0.7 (0.9) | 1.3 | 1.1 | 1.4 | 2283.0 | 15.1 | <.001 | 58.8 | 0.613 |
| (7) Win probability (%) | 21.8 (4.8) | 18.7 (8.4) | 3.1 | 2.5 | 3.7 | 1214.3 | 10.4 | <.001 | 70.8 | 0.424 |
| (8) Mean win (Euro) | 6.2 (10) | 1.8 (2.7) | 4.5 | 4.0 | 5.0 | 2262.4 | 17.5 | <.001 | 74.3 | 0.734 |
| (9) Median win (Euro) | 3 (5) | 1 (1.3) | 1.9 | 1.7 | 2.2 | 2248.0 | 15.2 | <.001 | 55.4 | 0.637 |
| (10) Mean loss (Euro) | 2.2 (3.2) | 0.7 (0.9) | 1.5 | 1.3 | 1.6 | 2345.8 | 18.2 | <.001 | 86.9 | 0.738 |
| (11) Median loss (Euro) | 1.9 (3.3) | 0.7 (0.8) | 1.3 | 1.1 | 1.4 | 2222.7 | 15.3 | <.001 | 59.2 | 0.620 |
| (12) Total spent (Euro) | 520.1 (2687.3) | 38.7 (152.5) | 481.4 | 356.9 | 605.9 | 1824.9 | 7.6 | <.001 | 11.3 | 0.308 |

Note: Parameter = behavioral indicators compared between the two groups. See the text for an explanation for each parameter. High, Low = means of parameters for the high- and low-involvement groups, with standard deviations in parentheses. diff = difference between the high-involvement group and the low-involvement group. lowerCI, upperCI = lower and upper boundary of 95% confidence intervals of the difference. df, t, p = degrees of freedom, t value and p value from the Welch's t-tests. P values were corrected for multiple comparisons using the Holm-Bonferroni method. lnBF = the natural logarithm of Bayes factors. g = Hedges's average g.

amount (in euro), in the rounds where players lost, and (12) the total amount of money spent per player (in euro; a positive value indicates that they overall lost money).

The high-involvement players played more sessions and rounds in total. Their sessions were longer (i.e., contained more rounds) on average. The high-involvement players bet more, and as a result won more money when they won a game and lost more money when they lost a game. The overall probability of winning was higher in the high- than the low-involvement group. Two not mutually exclusive explanations exist for the latter finding. First, some players might play relatively few games, and decide to stop after experiencing a low winning probability. These players were more likely to be classified into the low-involvement group. Second, the game involved a skill component, as players needed to decide where to place each column. The high-involvement players might have a better knowledge of how to place the columns, and therefore achieved higher winning probabilities. Due to the negative expected value of the game, both groups lost money though (i.e., the total money spent was positive). Furthermore, the high-involvement players in total lost more money.

Most of the behavioral indicators we examined here overlapped with the ones used by Gaming1 to assign involvement levels to the players. These results thus served as 'manipulation checks', to show that the high-involvement group was indeed more involved in *Mystery Arena* than the low-involvement group. Next, we examined the influence of gambling outcomes and players' involvement level on (1) when to stop, (2) change in stake, and (3) the speed of play.

## When to stop

The ANOVA on the likelihoods of stopping after winning and losing (controlling for the overall probability of ending a session) revealed a significant main effect of prior outcome (Table 4). Players were more likely to stop after a loss than after a win (Fig 3, Panel (A)). This result is thus opposite to the prediction of loss-chasing (Table 2), but consistent with the results by Leino and colleagues on EGMs players [9]. Players in the current data set thus did not seem to chase losses; instead, they chased wins, by being more likely to continue playing after winning.

**Table 4. Statistical analyses on when to stop.**

*Number of players and rounds included in the analyses in each group*

| Group | Player | Round (Mean) | Round (SD) | Round (Min) | Round (Max) |
|---|---|---|---|---|---|
| High-Involvement | 1679 | 6001.9 | 14176.8 | 12 | 200706 |
| Low-Involvement | 651 | 440.9 | 959.3 | 13 | 13114 |

*ANOVA*

| Effect | df | MSE | F | ges | p |
|---|---|---|---|---|---|
| Involvement Level (High vs. Low) | 1, 2328 | 0.09 | 5.36 | <.001 | .021 |
| Prior Outcome (Loss vs. Win) | 1, 2328 | 0.29 | 3353.74 | .520 | <.001 |
| Interaction | 1, 2328 | 0.29 | 8.95 | .003 | .003 |

*Pairwise comparisons*

| Comparison (A vs. B) | A-mean | B-mean | diff | lowerCI | upperCI | df | t | p | lnBF | g |
|---|---|---|---|---|---|---|---|---|---|---|
| High-Loss vs. High-Win | 1.24 (0.15) | 0.17 (0.49) | 1.07 | 1.04 | 1.10 | 1678.0 | 68.5 | <.001 | 1115.02 | 3.297 |
| Low-Loss vs. Low-Win | 1.21 (0.22) | 0.25 (0.79) | 0.96 | 0.88 | 1.04 | 650.0 | 24.4 | <.001 | 207.52 | 1.884 |
| High-Loss vs. Low-Loss | 1.24 (0.15) | 1.21 (0.22) | 0.03 | 0.01 | 0.05 | 898.1 | 3.0 | .007 | 3.33 | 0.140 |
| High-Win vs. Low-Win | 0.17 (0.49) | 0.25 (0.79) | -0.08 | -0.14 | -0.01 | 850.1 | -2.3 | 0.027 | 0.83 | 0.105 |
| (High-Loss—High-Win) vs. (Low-Loss—Low-Win) | 1.07 (0.64) | 0.96 (1.01) | 0.10 | 0.02 | 0.19 | 860.4 | 2.5 | 0.027 | 1.47 | 0.114 |

Note: *ANOVA*: df = degrees of freedom. In a 2 by 2 ANOVA, the dfs for all effects are the same. MSE = mean square of the error. ges = generalized eta squared. *Pairwise comparisons*: Comparison (A vs. B) = the two variables compared in each comparison. A-mean, B-mean = means of the left (A) and the right (B) variable in a comparison, with standard deviations in parentheses. diff = difference between A and B. lowerCI, upperCI = lower and upper boundary of 95% confidence intervals of the difference. df, t, p = degrees of freedom, t value and p value from the Welch's t tests (between-subjects comparisons) or paired-samples t tests (within-subjects comparisons). P values were corrected for multiple comparisons using the Holm-Bonferroni method. lnBF = the natural logarithm of Bayes factors. g = Hedges's average g.

Both the main effect of involvement level and the interaction effect were statistically significant. To break down these effects, we conducted pairwise comparisons. First, we compared the likelihood of stopping after losing versus winning for the two groups separately. Both groups were more likely to stop after a loss than after a win (High-Loss vs. High-Win and Low-Loss vs. Low-Win in Table 4), thus showing win-chasing in both groups. To examine if the extent of win-chasing differed between groups, for each player, we calculated a difference score between the likelihood of stopping after a loss and after a win. A positive difference score indicates win-chasing, with a larger value for a stronger tendency to chase wins. A direct between-group comparison showed larger difference scores for the high-involvement players ((High-Loss—High-Win) vs. (Low-Loss—Low-Win) in Table 4). The high-involvement players thus showed a stronger tendency to chase wins. Note though that the between-group difference is small (Hedges's g = 0.114), and the Bayes factor (lnBF = 1.47; BF = 4.35) only provided anecdotal evidence for the alternative hypothesis.

In this analysis, we relied on the operator data to define one session of play, which was from when gamblers logged into the platform till when they logged off. However, players may also take a break without logging off. In an extra analysis, we assumed that players had taken a break when they took more than 10 minutes to respond, and that a new session started after they took a break. Both groups were still more likely to stop after a loss than after a win. The tendency to chase wins was descriptively larger in the high-involvement group, but the effect size was again quite small (Hedges's g = 0.100) and the Bayes factor was inconclusive (lnBF = 0.39, BF = 1.48; see the Supplemental Materials). The between-group difference in the extent of win-chasing thus does not seem robust, and is at best small.

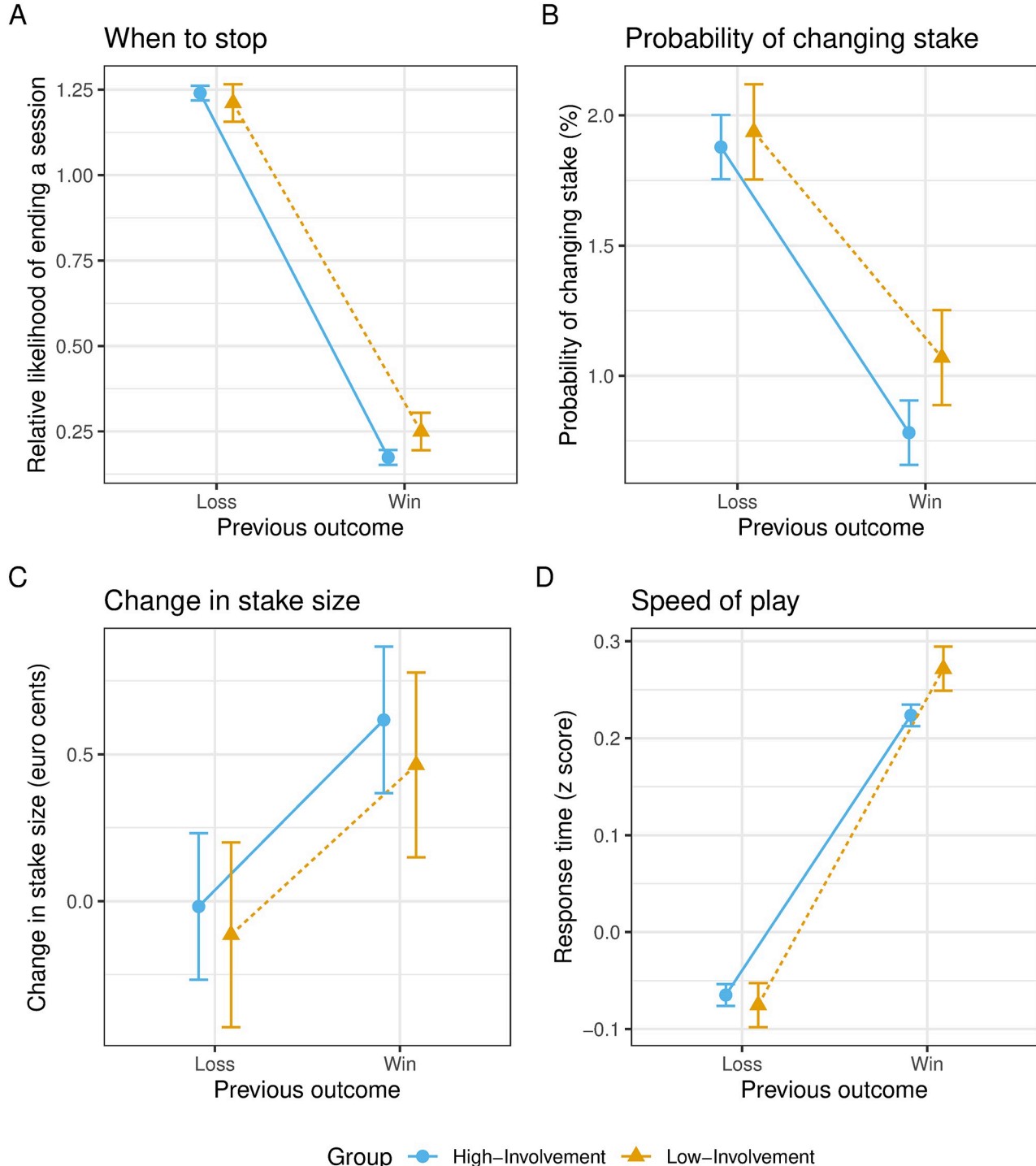

**Fig 3. Behavioral expressions of within-session chasing.** (A) when to stop, (B) probability of changing stake, (C) change in stake size, and (D) speed of play. For panel (A), relative likelihoods are the conditional probabilities of stopping after a loss and after a win, normalized per player by the overall probability to stop, e.g. p(*stop*|*loss*)/p(*stop* − *overall*) and p(*stop*|*win*)/p(*stop* − *overall*). Error bars stand for 95% within-subject confidence intervals.

**Table 5. Statistical analyses on change in stake.**

*Number of players and rounds included in the analyses in each group*

| Group | Player | Round (Mean) | Round (SD) | Round (Min) | Round (Max) |
|---|---|---|---|---|---|
| High-Involvement | 1678 | 5936.6 | 14046.5 | 11 | 197613 |
| Low-Involvement | 648 | 435.5 | 949.0 | 13 | 12959 |

*Probability of changing stakes (%)—ANOVA*

| Effect | df | MSE | F | ges | p |
|---|---|---|---|---|---|
| Involvement Level (High vs. Low) | 1, 2324 | 17.85 | 1.57 | <.001 | .210 |
| Prior Outcome (Loss vs. Win) | 1, 2324 | 6.35 | 141.80 | .016 | <.001 |
| Interaction | 1, 2324 | 6.35 | 1.95 | <.001 | .163 |

*Probability of changing stakes (%)—Pairwise comparisons*

| Comparison (A vs. B) | A-mean | B-mean | diff | lowerCI | upperCI | df | t | p | lnBF | g |
|---|---|---|---|---|---|---|---|---|---|---|
| High-Loss vs. High-Win | 1.88 (3.09) | 0.78 (3.96) | 1.10 | 0.92 | 1.27 | 1677.0 | 12.3 | <.001 | 68.96 | 0.311 |
| Low-Loss vs. Low-Win | 1.94 (3.15) | 1.07 (3.40) | 0.87 | 0.61 | 1.12 | 647.0 | 6.6 | <.001 | 17.83 | 0.264 |
| High-Loss vs. Low-Loss | 1.88 (3.09) | 1.94 (3.15) | -0.06 | -0.34 | 0.23 | 1154.7 | -0.4 | 1.000 | -2.88 | 0.019 |
| High-Win vs. Low-Win | 0.78 (3.96) | 1.07 (3.40) | -0.29 | -0.61 | 0.04 | 1359.3 | -1.7 | 0.726 | -1.63 | 0.081 |
| (High-Loss—High-Win) vs. (Low-Loss—Low-Win) | 1.10 (3.65) | 0.87 (3.35) | 0.23 | -0.08 | 0.54 | 1273.0 | 1.4 | 1.000 | -1.99 | 0.067 |

*Change in stake amount (euro cents)—ANOVA*

| Effect | df | MSE | F | ges | p |
|---|---|---|---|---|---|
| Involvement Level (High vs. Low) | 1, 2324 | 11.23 | 1.30 | <.001 | .255 |
| Prior Outcome (Loss vs. Win) | 1, 2324 | 24.22 | 14.21 | .004 | <.001 |
| Interaction | 1, 2324 | 24.22 | 0.03 | <.001 | .858 |

*Change in stake amount (euro cents)—Pairwise comparisons*

| Comparison (A vs. B) | A-mean | B-mean | diff | lowerCI | upperCI | df | t | p | lnBF | g |
|---|---|---|---|---|---|---|---|---|---|---|
| High-Loss vs. High-Win | -0.02 (3.76) | 0.62 (5.09) | -0.64 | -0.99 | -0.28 | 1677.0 | -3.5 | <.001 | 2.61 | 0.144 |
| Low-Loss vs. Low-Win | -0.12 (2.47) | 0.46 (4.17) | -0.58 | -1.02 | -0.13 | 647.0 | -2.6 | 0.110 | 0.10 | 0.174 |
| High-Loss vs. Low-Loss | -0.02 (3.76) | -0.12 (2.47) | 0.10 | -0.17 | 0.36 | 1770.6 | 0.7 | 1.000 | -2.78 | 0.033 |
| High-Win vs. Low-Win | 0.62 (5.09) | 0.46 (4.17) | 0.15 | -0.25 | 0.56 | 1424.3 | 0.7 | 1.000 | -2.72 | 0.035 |
| (High-Loss—High-Win) vs. (Low-Loss—Low-Win) | -0.64 (7.37) | -0.58 (5.77) | -0.06 | -0.62 | 0.51 | 1490.7 | -0.2 | 1.000 | -2.94 | 0.009 |

Note: *ANOVA*: df = degrees of freedom. In a 2 by 2 ANOVA, the dfs for all effects are the same. MSE = mean square of the error. ges = generalized eta squared. *Pairwise comparisons*: Comparison (A vs. B) = the two variables compared in each comparison. A-mean, B-mean = means of the left (A) and the right (B) variable in a comparison, with standard deviations in parentheses. diff = difference between A and B. lowerCI, upperCI = lower and upper boundary of 95% confidence intervals of the difference. df, t, p = degrees of freedom, t value and p value from the Welch's t tests (between-subjects comparisons) or paired-samples t tests (within-subjects comparisons). P values were corrected for multiple comparisons using the Holm-Bonferroni method. lnBF = the natural logarithm of Bayes factors. g = Hedges's average g.

## Change in stake

Players overall changed stake very infrequently (Fig 3, Panel (B)). Only the main effect of prior outcome was statistically significant (Table 5). Players were more likely to change stake after a loss than after a win, similar to the *win-stay, lose-shift* strategy often observed in competitive games [47]. This effect was observed in both groups (High-Loss vs. High-Win and Low-Loss vs. Low-Win for probability of changing stakes in Table 5). For the between-group comparisons, no statistically significant effects were found, and the BFs all provided moderate to strong support for the null effect (Table 5).

For the change in stake amount (in euro cents), again only the main effect of prior outcome was statistically significant (Table 5). Players on average increased their stake after a win, but not after a loss (Fig 3, Panel (C)). As a result, the increase in stake amount was larger after a win than after a loss. Players were thus less likely to change stake after winning, but when they

did, they tended to increase the stake. The same pattern was observed for both groups. However, the effect was not statistically significant for the low-involvement group after correcting for multiple comparisons, and the BF was inconclusive (Low-Loss vs. Low-Win in Table 5 for change in stake amount; lnBF = 0.10, BF = 1.11). The between-group comparisons all yielded strong support for the null effect. Overall, players did not chase losses in stake amount. If anything, they showed a small tendency to chase wins, by betting a slightly larger stake after winning.

## Speed of play

As mentioned above, the high-involvement players overall played more quickly than low-involvement players (see the Supplemental Materials for an analysis on raw RTs). We therefore focused on the RT z scores here. The main effect of involvement level, the main effect of prior outcome and their interaction effect were all statistically significant (Table 6). Players played more quickly after a loss than after a win (Fig 3, Panel (D)), and this effect was observed for both high- and low-involvement players (High-Loss vs. High-Win and Low-Loss vs. Low-Win in Table 6). Both groups thus showed loss-chasing in their speed of play. To examine whether the extent of loss-chasing differed between the two groups, for each player, we calculated a difference score between RT z scores after a loss and after a win (i.e., $RT_{diff} = RT_{loss} - RT_{win}$). A negative difference score indicates a tendency to chase losses, with more negative values for stronger tendencies to chase losses. These difference scores were then compared between the two groups. We found that the difference scores were *less* negative in the high-involvement group than in the low-involvement group ((High-Loss—High-Win) vs. (Low-Loss—Low-Win) in Table 6), suggesting that gambling outcomes had a smaller influence on the speed of

**Table 6. Statistical analyses on speed of play.**

*Number of players and rounds included in the analyses in each group*

| Group | Player | Round (Mean) | Round (SD) | Round (Min) | Round (Max) |
|---|---|---|---|---|---|
| High-Involvement | 1678 | 5914.4 | 13975.3 | 11 | 196747 |
| Low-Involvement | 646 | 433.3 | 943.7 | 16 | 12856 |

*ANOVA*

| Effect | df | MSE | F | ges | p |
|---|---|---|---|---|---|
| Involvement Level (High vs. Low) | 1, 2322 | 0.02 | 15.22 | .002 | <.001 |
| Prior Outcome (Loss vs. Win) | 1, 2322 | 0.06 | 1483.85 | .322 | <.001 |
| Interaction | 1, 2322 | 0.06 | 12.55 | .004 | <.001 |

*Pairwise comparisons*

| Comparison (A vs. B) | A-mean | B-mean | diff | lowerCI | upperCI | df | t | p | lnBF | g |
|---|---|---|---|---|---|---|---|---|---|---|
| High-Loss vs. High-Win | -0.065 (0.074) | 0.224 (0.259) | -0.288 | -0.304 | -0.273 | 1677.0 | -35.8 | <.001 | 472.22 | 1.732 |
| Low-Loss vs. Low-Win | -0.075 (0.090) | 0.272 (0.331) | -0.347 | -0.379 | -0.315 | 645.0 | -21.1 | <.001 | 166.10 | 1.643 |
| High-Loss vs. Low-Loss | -0.065 (0.074) | -0.075 (0.090) | 0.010 | 0.003 | 0.018 | 994.0 | 2.6 | 0.009 | 1.04 | 0.120 |
| High-Win vs. Low-Win | 0.224 (0.259) | 0.272 (0.331) | -0.048 | -0.076 | -0.020 | 963.6 | -3.3 | 0.003 | 3.79 | 0.153 |
| (High-Loss—High-Win) vs. (Low-Loss—Low-Win) | -0.288 (0.330) | -0.347 (0.417) | 0.058 | 0.023 | 0.094 | 971.5 | 3.2 | 0.003 | 3.25 | 0.148 |

Note: *ANOVA*: df = degrees of freedom. In a 2 by 2 ANOVA, the dfs for all effects are the same. MSE = mean square of the error. ges = generalized eta squared. *Pairwise comparisons*: Comparison (A vs. B) = the two variables compared in each comparison. A-mean, B-mean = means of the left (A) and the right (B) variable in a comparison, with standard deviations in parentheses. diff = difference between A and B. lowerCI, upperCI = lower and upper boundary of 95% confidence intervals of the difference. df, t, p = degrees of freedom, t value and p value from the Welch's t tests (between-subjects comparisons) or paired-samples t tests (within-subjects comparisons). P values were corrected for multiple comparisons using the Holm-Bonferroni method. lnBF = the natural logarithm of Bayes factors. g = Hedges's average g.

play for high-involvement players. In other words, the high-involvement players showed a smaller loss-chasing effect in their speed of play.

## Discussion

Using data unobtrusively recorded while players played the online gambling product *Mystery Area*, we examined three behavioral expressions of within-session loss-chasing. Overall, both high- and low-involvement players were less likely to stop after a win than after a loss (i.e., win-chasing in when to stop). Furthermore, both groups increased the stake amount more after a win than after a loss (i.e., win-chasing in stake amount). For the speed of play, both groups responded more quickly after a loss than after a win (i.e., loss-chasing in speed of play), with a smaller effect in high-involvement players. Below we first discuss each behavioral expression of chasing in turn, and then discuss what these results imply for chasing in real gambling more generally.

### When to stop

To examine when players ended a session, we compared the probability of stopping after a win and after a loss, while controlling for the overall stopping probability for each player. In line with the finding by Leino et al. [9], we found no loss chasing in this facet. Players were more likely to continue playing after winning, thus showing win-chasing instead. Whether players can continue a gambling session or not is necessarily limited by the amount of money available. Winning provides gamblers with extra funds to continue, while after losing, they may have to stop due to insufficient funds. In other words, the fact that gamblers were more likely to stop after a loss does not necessarily mean that they voluntarily decided to stop. Instead, sometimes they might have had to stop because they ran out of money after losing. However, since we do not have data on the amount of money available in players' accounts, this hypothesis cannot be directly tested. This possibility nevertheless suggests that chasing losses by continuing playing may be constrained by available funds in real gambling, and future work can consider this constraint, by for instance recording funds available in gamblers' accounts.

### Change in stake

Players were more likely to change stakes after a loss than after a win (although the overall probability of changing stakes was quite low), in line with the win-stay, lose-shift strategy [47]. Moreover, they increased the stake amount more after winning than losing. No loss-chasing was therefore observed in stake change. Multiple explanations may explain the absence of loss-chasing here. First, gambling funds may constrain how much money gamblers can bet. Winning may provide players with more funds to place a larger bet, whereas after losing, players might not have sufficient funds to increase the stake. Second, players might become more risk-taking after winning (e.g., the house money effect [17]), thus showing win-chasing in stake amount. Note that in this game, gamblers seemed to pick a certain amount and rarely changed the stake when playing. This game and the way it was played by most gamblers may hence not be ideal for assessing how gamblers adjust stakes after wins and losses.

### Speed of play

Gamblers played more quickly after a loss than after a win, thus showing loss-chasing in the speed of play. This finding is consistent with previous findings from laboratory tasks and simulated gambling [23, 25, 26], and serves as the first demonstration of this effect in real gambling in a large-scale study. Surprisingly, the highly involved players showed a smaller effect. They

bet more money on average, and consequently won more money when they won a gamble, and lost more money when they lost a gamble. Previous work has shown that individuals tended to pause longer when the size of obtained reward increased [22]. Furthermore, it has been speculated that the omission of larger wins (i.e. larger losses in our study) may be more frustrating [25]. Based on this, one would predict a larger difference between wins and losses for high-involvement players (larger win/loss amounts) compared with low-involvement players (smaller win/loss amounts). However, we observed the opposite. The difference in the speed of play after a win versus a loss consists of two effects: slowing down after winning (the post-reinforcement pause effect [48]), and speeding up after losing (the post-loss speeding effect [25]). Due to the lack of a neutral baseline, we cannot distinguish between these two effects here. Accordingly, two explanations (not mutually exclusive) may account for the smaller effect for highly involved gamblers: the high-involvement group may be less sensitive to wins, and/or they may be less sensitive to losses (e.g., [49, 50]). Future research may utilize simulated gambling tasks that include neutral outcomes (e.g., [25]), and examine how the two effects may be related to gamblers' level of involvement.

## Chasing: A multifaceted phenomenon

One strength of the current work is the simultaneous examination of three behavioral expressions of within-session loss-chasing. Among the three facets examined, we observed win-chasing in (1) when to stop and (2) change in stake amount, and loss-chasing in (3) the speed of play. Instead of chasing wins or losses in all aspects of gambling behavior, gamblers seemed to intensify some facets after winning, and other facets after losing. This finding highlights the multifaceted and dynamic nature of chasing. To understand chasing, it is therefore necessary to simultaneously characterize multiple aspects of gambling behavior. We note that the speed of play may be an important facet via which chasing can be expressed. As discussed above, after losing, gamblers may not have enough money to place a larger stake, and might even be forced to stop. These factors may contribute to the observed win-chasing in the first two facets. By contrast, as long as gamblers have sufficient funds to continue playing, the speed of play is not limited by prior outcomes. As such, the speed of play may serve as a 'pure' measure of gambler's urge to continue playing, since it is not constrained by available funds. The speed of play has received less attention than the other two facets so far. Our results suggest that it might be fruitful to consider the speed of play when examining chasing in real gambling.

Wins and losses had comparable effects in both the low- and high-involvement groups. Overall, we did not observe a stronger tendency to chase losses in players who were more involved in gambling. This finding seems to run counter to the idea that loss-chasing is a defining feature of problem gambling [6], and may thus be more pronounced as people become more involved in gambling. One possibility is that gambling involvement based on operator tracking data may not correlate with problem gambling severity (see below). Second, the clinical diagnosis of gambling disorder (e.g., DSM-5 [4]) uses between-session chasing as a criterion, while we examined within-session chasing here. On the one hand, within-session chasing can be a building block for between-session chasing [8]. For instance, extensive and intensified gambling within a session may be more likely to lead to substantial financial losses (due to the negative expected value of most gambling products), which may lead gamblers to come back another time to try to recoup previous losses (i.e., between-session chasing). On the other hand, between-session chasing does not necessarily mean that loss-chasing is also intensified within a session. Indeed, although the highly involved players gambled more intensively, wins and losses in the previous round had a smaller influence on their speed of play. This potential dissociation between within- and between-session chasing again highlights the multifaceted

nature of loss-chasing, and the necessity of carefully delineating different facets of chasing in real gambling.

## Limitations and future directions

One important limitation of the current research is that we analyzed data from one specific gambling product, *Mystery Arena*. The fast-paced nature of *Mystery Arena* may make it especially suitable for assessing certain aspects of loss-chasing (e.g., the speed of play) but less so for other aspects (e.g., the change in stake, as discussed above). In other gambling products, loss-chasing may be expressed differently. Moreover, different gambling products may attract different types of gamblers, who may differ in their chasing behavior (e.g., online poker vs. casino [30]). Further examining loss-chasing in other gambling products will illuminate the generalizability of the current findings.

Second, the gamblers in the current sample were assigned different involvement levels based on player tracking data, using a proprietary algorithm by Gaming1. Although some indicators used by Gaming1 showed some overlap with the clinical criteria for gambling disorder in DSM-5 (see the Supplemental Materials), it is unclear to what extent the involvement levels indeed correlate with (the risk levels of) problem gambling. One fruitful avenue for future research is to use more validated measures of problem gambling, and examine the correlations between problem gambling severity and behaviors tracked by the operators. Such information will allow for a more systematic examination of loss-chasing behavior across the whole spectrum of gambling involvement, from infrequent social gamblers to those who have problems with their gambling.

## Conclusion

To conclude, using a large data set collected from an online commercial gambling product *Mystery Arena*, we examined three behavioral expressions of within-session loss-chasing for gamblers with high and low levels of gambling involvement. Overall, players were less likely to stop and tended to increase the stake after a win (i.e., win-chasing). However, after a loss, they played the game more quickly, which may reflect an urge to continue gambling (loss-chasing in the speed of play). Furthermore, wins and losses had a smaller influence on the speed of play for highly involved gamblers, which might reflect their reduced sensitivity to wins and/or losses. These findings highlight the multifaceted and dynamic nature of chasing in real gambling. Future studies may examine loss-chasing in other gambling products and in people with gambling problems to evaluate the generalizability of the current findings.

## Supporting information

**S1 File. Online supplemental materials.**
(PDF)

## Acknowledgments

We thank Olivier Massiot and Florian Merchie from Gaming1 for their invaluable assistance with the current project.

## Author Contributions

**Conceptualization:** Zhang Chen, Xavier Noël, Frederick Verbruggen.

**Data curation:** Zhang Chen.

**Formal analysis:** Zhang Chen, Roos Arwen Doekemeijer.

**Funding acquisition:** Xavier Noël, Frederick Verbruggen.

**Investigation:** Zhang Chen, Xavier Noël.

**Methodology:** Zhang Chen, Frederick Verbruggen.

**Project administration:** Zhang Chen, Xavier Noël, Frederick Verbruggen.

**Resources:** Xavier Noël.

**Software:** Zhang Chen, Roos Arwen Doekemeijer.

**Supervision:** Frederick Verbruggen.

**Validation:** Zhang Chen, Roos Arwen Doekemeijer.

**Visualization:** Zhang Chen, Roos Arwen Doekemeijer.

**Writing – original draft:** Zhang Chen.

**Writing – review & editing:** Zhang Chen, Roos Arwen Doekemeijer, Xavier Noël, Frederick Verbruggen.

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
