## [Decision Letter · Decision Letter 0]

2 Mar 2022

PONE-D-21-35671Winning and losing in online gambling: Effects on within-session chasingPLOS ONE

Dear Dr. Chen,

Thank you for submitting your manuscript to PLOS ONE. After careful consideration, we feel that it has merit but does not fully meet PLOS ONE’s publication criteria as it currently stands. Therefore, we invite you to submit a revised version of the manuscript that addresses the points raised during the review process.

Thank you for your patience, I had issues to secure enough reviewers. I could receive reviews of the manuscript and read the manuscript before reading the reviews. I share reviewers opinions and I have additional comments. I would appreciate if the authors could address all the reviewers' comments and mine :

On the overall, the introduction and discussion are well written, but the material and statistical analysis need to be reworked. That’s one of the main issues of this article. The methods and statistical analysis seem to have been correctly done but the way it is explained is far too complex. In addition, the discussion lacks perspective and there is no clear message regarding the results of this study under the scope of the current literature.

Introduction:

Hypothesis: It seems there was no clear hypothesis, and it was more an exploratory study. I would appreciate further explanation on this matter.

Material and methods:

The game is extremely simple but when referring to the figure and the explanation of the game gave by the authors, this game seemed too complex. I think the authors need to improve the figure 1 (also please place figure 1 under the figure), and the game explanation. Figure 1 should allow reader to understand how the game works at first sight. Essential information could be provided first, followed by the additional information for further details

Measures:

Same as previous comments, please provide essential information first and then provide details. A summarizing table with measures, tests and associated hypothesis would be appreciated.

Statistical analysis:

The presentation of the statistical analysis is too complex, and readers will get lost very quickly. The authors need to provide the purpose of each of the analyses so the readers will understand easily. There are several other issues such as: Line 235, nothing is specified about the two groups; Line 250 how the High-involvement and low-involvement players group are defined? I couldn’t find anything in the method.

Results

Figure 3 is misleading with the shifted lines between groups and the vertical reading arrangement. In addition, I do not understand why the three behavioral expressions are presented within the same figure. The arrangement of the figure should be based on a horizontal reading pattern with A and B as the top left and top right sub figure and the C as the bottom left and bottom right sub figure.Please submit your revised manuscript by Apr 16 2022 11:59PM. If you will need more time than this to complete your revisions, please reply to this message or contact the journal office at plosone@plos.org. Please include the following items when submitting your revised manuscript:A rebuttal letter that responds to each point raised by the academic editor and reviewer(s). You should upload this letter as a separate file labeled 'Response to Reviewers'.A marked-up copy of your manuscript that highlights changes made to the original version. You should upload this as a separate file labeled 'Revised Manuscript with Track Changes'.An unmarked version of your revised paper without tracked changes. You should upload this as a separate file labeled 'Manuscript'.

We look forward to receiving your revised manuscript.

Kind regards,

Gaëtan Merlhiot

Academic Editor

PLOS ONE

Journal Requirements:

We thank Olivier Massiot and Florian Merchie from Gaming1 for their invaluable 554

assistance with the current project. This work was supported by an ERC Consolidator 555

grant (European Union’s Horizon 2020 research and innovation programme, grant 556

agreement No 769595) and Methusalem Project 01M00221 (Ghent University) awarded 557

to Frederick Verbruggen. Xavier No¨el is supported by the Fonds de la Recherche 558

Scientifique (F.R.S.-FNRS), Belgium

This work was supported by an ERC Consolidator grant (European Union’s Horizon 2020 research and innovation programme, grant agreement No 769595; https://erc.europa.eu/) and Methusalem Project 01M00221 (Ghent University; https://www.ugent.be/en/research/funding/bof/methusalem) awarded to Frederick Verbruggen. Xavier Noël is supported by the Fonds de la Recherche Scientifique (F.R.S.-FNRS; https://www.frs-fnrs.be/en/), Belgium and research funding from BAGO (Belgian Association of Gaming Operators; https://bago.be/). The funders had no role in study design, data collection and analysis, decision to publish, or preparation of the manuscript.

I have read the journal's policy and the authors of this manuscript have the following competing interests: XN received research funding from BAGO (Belgian Association of Gaming Operators) of which Gaming1 is affiliated. No honorarium was received. There was no editorial direction or censorship from Gaming1. ZC, RD, and FV declare that no competing interests exist. 

Reviewers' comments:

Reviewer's Responses to Questions

**Comments to the Author**

1. Is the manuscript technically sound, and do the data support the conclusions?

Reviewer #1: Yes

Reviewer #2: Yes

2. Has the statistical analysis been performed appropriately and rigorously? 

Reviewer #1: Yes

Reviewer #2: Yes

3. Have the authors made all data underlying the findings in their manuscript fully available?

Reviewer #1: Yes

Reviewer #2: No

4. Is the manuscript presented in an intelligible fashion and written in standard English?

Reviewer #1: Yes

Reviewer #2: No

5. Review Comments to the Author

Reviewer #1: I have reviewed this manuscript previously – and positively – for a different journal, and I would like to say what a pleasant surprise it is to see that most of my previous concerns have been addressed in this new submission.

This is an unusual and sophisticated piece of research using naturalistic gambling data from a specific online gambling game (‘Mystery Arena’) in a large sample (over 2500 gamblers making over 10 million bets). The paper provides a detailed behavioural analysis of ‘loss chasing’, a key feature of problematic gambling. An important contribution is the description of 3 distinct expressions of loss chasing: ‘when to stop’ (linked to persistence), the amount bet, and the speed of play. They compare these variables between winning and losing gambles, and between groups classified by the gambling operator as higher or lower risk, using their own (proprietary) algorithm. The indicators used by the gambling operator are shown in the Supp material, and while this classification is a little opaque (given the proprietary aspect), there is certainly useful information in these analyses, and I thought the authors handled its (limited) validity very appropriately. The groups differ primarily on the stopping variable (although not in the conventional ‘loss’ chasing direction) and on the speed of play variable.

As with much research, a key strength is also a key weakness: the authors focus on Mystery Arena as one specific gambling product, because chasing may be expressed differently on different kinds of gambling (e.g. slot machine gamblers may persist whereas sports bettors may escalate their stake size). The authors also note that it is better suited to the speed of play analysis that a conventional slots game. I fully agree on both accounts, but I might add as a limitation that it seems a slightly unorthodox game. The authors refer on pg 16 that it is ‘chance based’ and compare on pg 4 to a ‘conventional slot machine’. I would say that the behavioural requirements per bet (‘skill’, in a colloquial sense) are greater than a slot machine, and in the manipulation checks, the high involvement do seem to win more than the low involvement groups. To what extent do the motor requirements of the game actually afford experienced players control over the return to player? Could the authors add a clear statement (perhaps in the task methods) as to where this game falls on the skill /chance continuum? Later on in the manuscript, they also refer to the game as a “gamble” e.g. Discussion line 1. I would suggest term ‘gambling product’.

Of the 3 definitions of chasing, I feel the ‘when to stop’ analysis is the most challenging; for the change in stake and speed of play analyses, I feel there would be much agreement among researchers about how to operationalize those variables. For ‘when to stop’, the authors approach this in an interesting way, by looking at the probability of winning for that player’s overall ‘session’, and the pwin on their last trial (i.e. before they stop). As p(win-end) is less than p(win-overall), they conclude that players stop when they are losing, i.e. the opposite of loss ‘chasing’. This is the first paper that I have seen operationalize chasing in this way, and it’s an interesting way of simplifying a very complex variable. At the same time, it loses some of the dynamics of chasing, by ignoring the length of the session, and it doesn’t seem to adequately orthogonalize the responses to winning and losing. For example, the authors acknowledge that their effect (the low pwin-end) could reflect stopping after a losing streak, or bigger wins early in the session, and/or exhausting funds. The secondary analysis excluding breaks over 10 minutes does not really shed any light on these ambiguities. Overall, I think the authors could stick with this analysis, but I would encourage them to note that there are other ways to operationalize this variable, e.g. the logistic regression by Leino.

Minor

P2 please use ‘person first language’ i.e. people with gambling problems, not ‘problem gamblers’

Pg 6 line 218 – 230. Rather than endorsing or rejecting the terminology from the gambling operator (‘pathological addictive gambler’ etc), I think this section could be shortened just by introducing the five levels and the binarization.

Table 1 please add units to the parameters, e.g. is the mean and median win in Euros?

Not a huge problem but printed in black and white, the two lines in Fig 3 and elsewhere are indistinguishable.

Reviewer #2: In this manuscript the authors study the phenomenon of “loss chasing” in gambling data from a large dataset obtained from an online gambling company. The experimental setting describe an agent engaged with a repeated decision-making task, where in each trial the agent may choose to continue or halt participation in a gamble that may result a win or a loss. If the agent decides to continue gambling, they also choose whether to change the previous gambled amount (the “stake”). In this framework, loss chasing is the tendency to increase the stakes and probability of choosing to continue gambling specifically in trials that follow losses. In terms of the probability to continue gambling, the authors identify the opposite of loss chasing with an increased likelihood to stop gambling after losing compared to after winning (Fig 3.a). In changing gambled stakes, participants were more likely to change their stake after a loss compared to a win. However, stake change after loss and win were both very rare (2% and 1% to change stake respectively), and the amount of stake changing after loss was centered around zero (Fig 3.b). In testing reaction times, the authors found that participants made quicker decisions following a loss compared to wins. Across these behavioral features, the authors compare the behavior of light and heavy gamblers.

The manuscript seems methodologically sound. Given the importance of better understanding gambling behavior, and the limitation in studying such behavior in the lab setting, the extensive ecological dataset analyzed here is valuable in unveiling the underlying mechanisms of gambling in its common and pathological forms. My main concern in reviewing the manuscript is a lack of clear message. Having studied the data, what new insights did we gain about gambling behavior? Does the evidence support the notion of loss chasing or does it not? What do the findings suggest on the decision-making mechanisms of gamblers? What do they imply to the vulnerability of gamblers to specific types of games? How would these vulnerabilities be expressed in different types of gambling games? How could the novel understanding from the present study be utilized for the design of interventions in gambling disorders? It is not necessary to address all these questions. Rather, a better distilling of a main message and its implications in some broader context could strengthen the manuscript and its impact, and make it easier to follow. Minor refinements or a reframing of the introduction, and some expansion of the discussion may accomplish this goal.

Major comments

-------------------

* The main effect of loss chasing is implicated in the tendency to chase, namely continue gambling (possibly with greater stakes), after a loss. As the authors suggest, comparing the groups of high- and low-involvement gamblers should consider each group’s winning base-rate, which is higher for the high-involvement groups. This is equivalent to the comparison of two geometrical distributions with a different p (“The probability distribution of the number X of Bernoulli [with probability p] trials needed to get one success” [Wikipedia]). The question of whether participants engage in loss chasing is equivalent to asking what is the probability, given the data, that participants do not in fact have a fixed probability p for stopping, but that p is conditioned on the outcome of the previous trial. Figure 3.a greatly demonstrate this effect. Still, because it was unintuitive for me to mentally compare two geometrical distributions with different p’s, it was difficult for me to appreciate the magnitude of this effect in light of a null model which assumes that participants chose to stop gambling regardless of the last outcome. One way to help the reader could be to simulate the null model and position the present data within this simulation. One way to approach this is to simulate a vector of wins and losses according to the empirical distribution of probabilities to win or lose in the studied game. With this simulated vector of wins and losses, consider the probabilities pStop_loss^high, pStop_win^high, pStop_loss^low, pStop_win^low as the simulated probability to stop after a loss or a win in the high and low involvement groups. A null model would assume that pStop_win=pStop_loss= the empirical probability to stop. The distribution of probability of to stop for each of the groups may serve as the null hypothesis which by comparison would allow to appreciate the current data. After simulating this null hypothesis, subsequent simulations may push pStop_win and pStop_loss gradually apart (pStop_loss-pStop_win=0.01, 0.02,…). The resulting sequence of distributions (per each [pStop_loss-pStop_win] difference) may allow the estimation of: a. what is the most probable p_win and p_loss given the data, b. how likely it is that they are different within each group and c. between groups.

Minor comments

-------------------

* The introduction is somewhat repetitive and could be edited to be shorter and more concise. In general, I could easily understand the text, but the manuscript could benefit from further editing and proofreading.

* It might not be my place to comment on this, but it stood out - Given that one of the perspectives of the manuscript is gambling disorder, it is slightly odd that the first sentence of the introduction frame gambling as a recreational activity that people engage with for pleasure and fun (in the same way that a paper on endangered species may choose to refrain from framing sports hunting as a recreational activity that people engage with for pleasure; even if evidently both claims are true).

* “The influential pathways model proposed three distinct pathways from initial 8 gambling exposure to problem gambling [5] While different psychological and biological 9 processes are involved in the distinct pathways, all three …” It is not clear to the reader who is unfamiliar with the pathway model that problem gambling is one type of a pathway. More broadly, if the three different paths are of interest (it seems they are), it would assist the reader if they would be clearly defined here.

* line 14: “It is also one of the only signs of disordered gambling that can be directly captured in gambling behavior… The decision to continue gambling despite substantial financial losses is arguably the 38 most prototypical behavioral manifestation of loss-chasing” It is not essentials to your claims, but the reader remains wondering – why is this true? Is this a clinical, mechanistic, or other type of argument?

* Figure 1 has no caption, and I think a caption would assist the reader.

* Figure 1 shows the conversion between points and prizes which for most of the table has a conversion factor of 50 (100 points equal a prize of 20.0), but for the higher amounts of 600 and 1000 the conversions rate changes. I couldn’t understand if this is on purpose and of importance (it may be addressed in the text and I just missed it).

* It would assist the understanding of the game if several game-strategies could be explained or demonstrated. It seems there are a few degrees of freedom to participants of the game – where to place the columns, when to stop. Is the game based purely on luck (e.g. like a roulette) or does it have some elements of competency (like black jack)? Namely, could the game be played better by a competent player?

* Also, it wasn’t completely clear how are the cost of participating in a game calculated – are they determined per new draw of a dice-column? If so, could the stakes be modified between draws of dices? Does it mean that costs increased with each decision to draw a new column? Or, was participation cost a single amount set prior to a session? If so, why would it be beneficial to stop before the end of a session (before filling all the columns)?

* A related question is how did the game itself operate – were dices randomly drawn at each round with uniform distribution? Or by a different distribution? Was the game reactive (changed dice allocation based on the user choices)?

* This is a matter of taste: Figures 2 displays the distribution of age and gender across the two groups. This figure does not, of course, hinder understanding, but age and gender statistics seem like the least interesting aspect of the data; visualizing any of the rows of table 1 would be, subjectively for me, more relevant to the substance of the manuscript.

* Although it appears elsewhere, it would be helpful to add the number of participants per group to table 1 (e.g., to remind the reader that there were many more “High” participants compared to “low” ones; thus, explaining how come their “Round number” is so much higher).

* In table 2, 3, 4 the ANOVA’s df are identical to the main effects and interaction. I guess this is by design, but it is unintuitive (ANOVA tables for factors A and B are often presented with df |A|-1, |B|-1, (|A|-1)(|B|-1)). A comment addressing this would have helped me understanding the tables.

* Table 2 – I couldn’t understand the label of the last row. Is it “High- vs. Low-Involvement (Difference[Session End, session Overall])” what is “left” and what is “right” here?

* The Introduction is somewhat confusing in that when I approached the results, I was under the impression that the authors are about to demonstrate loss chasing in its most intuitive form – that gamblers tend to chase (continue gambling) after losses. The fact that the data suggests the opposite is important and interesting; but the expectations build up I had given previous sections left my quite confused. The way I read it, I was expecting to find that losses were followed by increased tendency to stop, and then the respective result is just glossed over (“Overall, p(win-end) was significantly lower than p(win-overall) 350 (Fig 3, Panel (A)). Players were thus more likely to stop after losing than after winning”).

In short, the way the introduction is written I was expecting to find that participants have greater likelihood to gamble after losses and it took me some time to understand that what the authors find is the opposite. It would have been easier for me to comprehend the text if the introduction would “prepare” me to this result, or be more clear about the fact that this result is surprising; or more explicit that this findings is inconsistent with loss chasing.

6. PLOS authors have the option to publish the peer review history of their article (what does this mean?). If published, this will include your full peer review and any attached files.

Reviewer #1: **Yes: **Luke Clark

Reviewer #2: **Yes: **Ohad Dan, Ph.D.

---

## [Author Response · Author response to Decision Letter 0]

11 May 2022

Please see the attached response to reviewers letter to find our responses to the comments by the editor and the reviewers.

---

## [Decision Letter · Decision Letter 1]

13 Jul 2022

PONE-D-21-35671R1Winning and losing in online gambling: Effects on within-session chasingPLOS ONE

Dear Dr. Chen,

Thank you for submitting your manuscript to PLOS ONE. After careful consideration, we feel that it has merit but does not fully meet PLOS ONE’s publication criteria as it currently stands. Therefore, we invite you to submit a revised version of the manuscript that addresses the points raised during the review process.

I am glad to see that all reviewers’ comments were considered. Please revise the manuscript to address the remaining minor concerns according to reviewers’ suggestions.

For Lab, Study and Registered Report Protocols: These article types are not expected to include results but may include pilot data.  Please submit your revised manuscript by Aug 27 2022 11:59PM. If you will need more time than this to complete your revisions, please reply to this message or contact the journal office at plosone@plos.org. Please include the following items when submitting your revised manuscript:A rebuttal letter that responds to each point raised by the academic editor and reviewer(s). You should upload this letter as a separate file labeled 'Response to Reviewers'.A marked-up copy of your manuscript that highlights changes made to the original version. You should upload this as a separate file labeled 'Revised Manuscript with Track Changes'.An unmarked version of your revised paper without tracked changes. You should upload this as a separate file labeled 'Manuscript'.If applicable, we recommend that you deposit your laboratory protocols in protocols.io to enhance the reproducibility of your results. Protocols.io assigns your protocol its own identifier (DOI) so that it can be cited independently in the future. For instructions see: https://journals.plos.org/plosone/s/submission-guidelines#loc-laboratory-protocols. Additionally, PLOS ONE offers an option for publishing peer-reviewed Lab Protocol articles, which describe protocols hosted on protocols.io. Read more information on sharing protocols at https://plos.org/protocols?utm_medium=editorial-email&utm_source=authorletters&utm_campaign=protocols.

We look forward to receiving your revised manuscript.

Kind regards,

Gaëtan Merlhiot

Academic Editor

PLOS ONE

Journal Requirements:

Reviewers' comments:

Reviewer's Responses to Questions

**Comments to the Author**

1. If the authors have adequately addressed your comments raised in a previous round of review and you feel that this manuscript is now acceptable for publication, you may indicate that here to bypass the “Comments to the Author” section, enter your conflict of interest statement in the “Confidential to Editor” section, and submit your "Accept" recommendation.

Reviewer #1: All comments have been addressed

Reviewer #2: All comments have been addressed

2. Is the manuscript technically sound, and do the data support the conclusions?

Reviewer #1: Yes

Reviewer #2: Yes

3. Has the statistical analysis been performed appropriately and rigorously? 

Reviewer #1: Yes

Reviewer #2: Yes

4. Have the authors made all data underlying the findings in their manuscript fully available?

Reviewer #1: Yes

Reviewer #2: Yes

5. Is the manuscript presented in an intelligible fashion and written in standard English?

Reviewer #1: Yes

Reviewer #2: Yes

6. Review Comments to the Author

Reviewer #1: The authors have been very attentive to the referee comments. This is a sophisticated piece of research: they are using high-resolution behavioural gambling data from a real online gambling platform, in a large dataset (10 million bets), looking at three distinct behavioural expressions that each require careful operationalizing. The value of the paper is in the contrast of the three expressions of chasing. They have done an excellent job of conveying their objectives and analysis pipeline as clearly as possible.

Reviewer #2: I was very happy to receive the updated manuscript. The revised version substantially improves on its previous. The Introduction is more concise, the Methods easier to follow, and the discussion better explains the multifaceted nature of loss chasing. In particular, I found table 1 to be helpful in shaping the reader’s expectations from the findings. Similarly, the grouping of all findings into a single figure (Fig. 3) is helpful. I thank the authors for the comprehensive point-by-point responses to my previous comments.

I have only minor comments:

1. Line 153 - “Note that players could also start a new round before the ’PLAY AGAIN’ message appeared” – I wasn’t sure what to make of this comment. If I understand correctly, games that concluded in a win were followed by a pleasant animation that participants may have chosen to savor. Games that were concluded by a loss were not. The comparison of timing between [win, something nice, continue?] and [lose, continue?] suggests then that the choice to continue to an additional game would be – by the construction of the game – quicker after losses, as indeed indicated by the results (e.g., Fig 3.D). Is this comment offered as a limitation to the finding of quicker RT after a loss? If so, it can potentially be framed as such more explicitly. Also, a comparison of the animation time and the difference between wins and losses reaction time may shed light on the role of the animation in generating this effect (if the animation is 1 second and the mean difference between wins and losses RT is 5 seconds then the effects of the animation are potentially of lesser concern). Alternatively, does this comment just explains the actual structure of the game and has no relation to RT?

2. I do not understand participants inclusion criterion. Line (236): “players needed to have at least 5 wins and 5 losses in the data”, line (255) “players again needed to have at least 5 rounds following a win and 5 rounds following a loss” – the ‘again’ seems to imply this is the same criterion but I’m not sure that it is: a player may have a single loss followed by 6 wins, satisfying the second condition but not the first.

3. Fig 3.a axis labels – The use of relative likelihoods of ending a session is well explained. However, I don’t know “relative likelihood” as a standard term. Due to the importance of the panel, and the possibly not so intuitive units of that panel, it would have been easier for me if “relative likelihood” was also described in the figure caption. Something like “relative likelihoods are the conditional stopping probabilities given a win or a loss, normalized per-participant by the overall probability to stop, e.g. for losses P(stop|loss)/P(stop-overall)”.

7. PLOS authors have the option to publish the peer review history of their article (what does this mean?). If published, this will include your full peer review and any attached files.

Reviewer #1: **Yes: **Luke Clark

Reviewer #2: **Yes: **Ohad Dan

---

## [Author Response · Author response to Decision Letter 1]

14 Jul 2022

Dear editor,

 Thank you again for the opportunity to revise our manuscript. In this letter, we list the comments by the two reviewers in bold, and explain how we have addressed each comment by Reviewer #2. Texts from the revised manuscript are shown in green. In the pdf file of the revised manuscript, we have highlighted the changes.

Kind regards,

Zhang Chen

Reviewer #1: The authors have been very attentive to the referee comments. This is a sophisticated piece of research: they are using high-resolution behavioural gambling data from a real online gambling platform, in a large dataset (10 million bets), looking at three distinct behavioural expressions that each require careful operationalizing. The value of the paper is in the contrast of the three expressions of chasing. They have done an excellent job of conveying their objectives and analysis pipeline as clearly as possible.

We would like to thank Reviewer #1 for these encouraging words, and for his constructive comments on our manuscript.

Reviewer #2: I was very happy to receive the updated manuscript. The revised version substantially improves on its previous. The Introduction is more concise, the Methods easier to follow, and the discussion better explains the multifaceted nature of loss chasing. In particular, I found table 1 to be helpful in shaping the reader’s expectations from the findings. Similarly, the grouping of all findings into a single figure (Fig. 3) is helpful. I thank the authors for the comprehensive point-by-point responses to my previous comments.

We would also like to thank Reviewer #2 for the constructive comments, which have greatly improved the quality and clarity of our manuscript.

I have only minor comments:

1. Line 153 - “Note that players could also start a new round before the ’PLAY AGAIN’ message appeared” – I wasn’t sure what to make of this comment. If I understand correctly, games that concluded in a win were followed by a pleasant animation that participants may have chosen to savor. Games that were concluded by a loss were not. The comparison of timing between [win, something nice, continue?] and [lose, continue?] suggests then that the choice to continue to an additional game would be – by the construction of the game – quicker after losses, as indeed indicated by the results (e.g., Fig 3.D). Is this comment offered as a limitation to the finding of quicker RT after a loss? If so, it can potentially be framed as such more explicitly. Also, a comparison of the animation time and the difference between wins and losses reaction time may shed light on the role of the animation in generating this effect (if the animation is 1 second and the mean difference between wins and losses RT is 5 seconds then the effects of the animation are potentially of lesser concern). Alternatively, does this comment just explains the actual structure of the game and has no relation to RT?

The feedback event is indeed generally longer after a win than after a loss. For this reason, we did not use the speed of initiating a new game as the indicator for the speed of play, but rather the interval between starting a new game till inserting the first column in a new game. The RT that we used is therefore not confounded by the factors mentioned by the reviewer, as we described in the data analysis section (line 271):

The speed of starting a round in the behavioral tracking data is the duration from when players placed the last column in the previous round till when they started the current round (Table 1). This duration included the different feedback events after winning and losing, and the time spent on adjusting the stake. Due to these confounding factors, we used the response time (RT) of placing the first column as the behavioral indicator of speed of play instead. This RT is from when players started the current round till when they put the first column in one of the 4 slots, and thus is not confounded by the factors mentioned above.

To make sure that readers clearly understand which RT indicator was used, when explaining the general procedure, we now immediately add a clarification in the revised manuscript (line 153):

Due to the differences in the feedback after wins and losses, and the flexibility in when to start a new round, we did not use the speed of starting a new round as the indicator for the speed of play. Instead, we used the interval between starting a new round till placing the first column in the new round (see Fig 1), which is not confounded by the factors discussed above.

2. I do not understand participants inclusion criterion. Line (236): “players needed to have at least 5 wins and 5 losses in the data”, line (255) “players again needed to have at least 5 rounds following a win and 5 rounds following a loss” – the ‘again’ seems to imply this is the same criterion but I’m not sure that it is: a player may have a single loss followed by 6 wins, satisfying the second condition but not the first.

The reviewer is correct in noting that these two inclusion criteria are not the same. To avoid misunderstanding, we have deleted the word ‘again’ on line 255.

3. Fig 3.a axis labels – The use of relative likelihoods of ending a session is well explained. However, I don’t know “relative likelihood” as a standard term. Due to the importance of the panel, and the possibly not so intuitive units of that panel, it would have been easier for me if “relative likelihood” was also described in the figure caption. Something like “relative likelihoods are the conditional stopping probabilities given a win or a loss, normalized per-participant by the overall probability to stop, e.g. for losses P(stop|loss)/P(stop-overall)”.

Thank you for this suggestion. We have now added an explanation on relative likelihoods in the caption of Figure 3.

For panel (A), relative likelihoods are the conditional probabilities of stopping after a loss and

after a win, normalized per player by the overall probability to stop, e.g. p(stop|loss)/p(stop − overall) and p(stop|win)/p(stop − overall).

---

## [Editor Report · Decision Letter 2]

8 Aug 2022

Winning and losing in online gambling: Effects on within-session chasing

PONE-D-21-35671R2

Dear Dr. Chen,

We’re pleased to inform you that your manuscript has been judged scientifically suitable for publication and will be formally accepted for publication once it meets all outstanding technical requirements.

Kind regards,

Gaëtan Merlhiot

Academic Editor

PLOS ONE
---

## [Editor Report · Acceptance letter]

10 Aug 2022

PONE-D-21-35671R2 

Winning and losing in online gambling: Effects on within-session chasing 

Dear Dr. Chen:

I'm pleased to inform you that your manuscript has been deemed suitable for publication in PLOS ONE. Congratulations! Your manuscript is now with our production department. 

Kind regards, 

on behalf of

Dr. Gaëtan Merlhiot 

Academic Editor

PLOS ONE